

# Restoring South African subtropical succulent thicket using *Portulacaria afra*: rooting variation across three soil types

Alastair J. Potts[1], Duncan Liddell[1], Catherine E. Clarke[2] and Nicholas C. Galuszynski[1]

[1] Department of Botany, Nelson Mandela University, Gqeberha, Eastern Cape, South Africa
[2] Department of Soil Science, University of Stellenbosch, Stellenbosch, Western Cape, South Africa

Corresponding author
Alastair J. Potts, potts.a@gmail.com

## ABSTRACT

**Aim:** Localised variation in soil properties can play an important role in shaping vegetation structure and plant community composition. However, in degraded ecosystems, these vegetation patterns may not be apparent due to the homogenization of local plant communities. Thus, defining restoration targets may prove challenging. By comparing the root development of *Portulacaria afra* (L) Jacq. cuttings grown in three different soils collected from degraded subtropical succulent thicket habitats, we aim to test whether soil chemical properties act as an ecological filter limiting root growth, which may in turn influence community composition and restoration success. This study focuses on root biomass differences as a proxy for potential vegetation structure rather than directly assessing plant community composition.

**Location:** Eastern Cape, South Africa.

**Taxon:** *Portulacaria afra* (L) Jacq.

**Methods:** Soil was collected from the top 20 cm at three sites within a degraded succulent thicket landscape (two with historically closed canopies, and one with a historically open woody canopy). A total of 196 *P. afra* cuttings were grown in each soil (588 cuttings in total) across two growing conditions (glasshouse and growth chamber). Root development was evaluated by destructive harvesting of 14 cuttings per treatment (soil type and growth condition) per week, and root dry weight was compared across treatments for each harvest date. The soil properties from each site were analysed to identify possible drivers for any differences in root development and visualised *via* a principal components analysis.

**Results:** Significant differences in root dry weight were detected (all tests: $F_{5,74} = 4.11$–$11.45$, $p < 0.01$). Root biomass was significantly lower in cuttings grown in soil from Site C (calcareous; historically open canopy) compared to Sites A and B (slightly saline and non-saline, respectively; historically closed-canopy thicket), suggesting that edaphic factors may have historically influenced vegetation structure. The soil from Site C showed notable differences from the soils at Sites A and B, with a higher pH (7.9 *vs* 6.5, 6.8, respectively), increased $Ca^{2+}$ concentration (25.4 *vs* 8.8, 6.4 cmol(+)/kg), Ca saturation % (83 *vs* 62, 53), and a lower P concentration (<2.2 *vs* 116, 43 mg/kg). These factors, particularly high pH and low P availability, likely suppressed root initiation and development, which may limit *P. afra* establishment in restoration efforts on calcareous soils.
**Conclusion:** Local variation in soil properties plays an important role in the regeneration dynamics and restoration of succulent thicket vegetation. Calcareous soils likely supported an open canopy vegetation with relatively low *P. afra* cover. This possible vegetation structure should be accounted for when setting restoration targets and measuring restoration success.

## INTRODUCTION

The spatial distribution of plant species within an ecosystem is a product of environmental filters operating at various spatial scales. Regional climate patterns often determine the broad vegetation types of an area (*Ohmann & Spies, 1998*; *Siefert et al., 2012*), whereas edaphic conditions can shape plant communities at local scales within vegetation types (*Acebes et al., 2010*; *Moro et al., 2015*; *Ohmann & Spies, 1998*; *Siefert et al., 2012*). The filtering of species based on local soil conditions does not necessarily represent a complete turnover of species, but rather a shift in community composition or structure (*Acebes et al., 2010* and references therein). In degraded ecosystems, the impact of anthropogenic activities outweighs the role of local environmental filters and homogenises local plant communities. Therefore, selecting the correct target areas for plant reintroductions may require identifying environmental filters, such as soil types. Here, we explore the effect of local soil conditions on the establishment of *Portulacaria afra* Jac., the target plant species used restoring severely degraded subtropical succulent thicket vegetation endemic to South Africa (*Mills et al., 2015*).

Subtropical succulent thicket (*sensu* the arid forms of the Albany Thicket biome; *Dayaram et al., 2019*; *Hoare et al., 2006*) occurs across a broad range of edaphic conditions where there is sufficient moisture and protection against disturbance (*Cowling & Potts, 2015*; *Vlok, Euston-Brown & Cowling, 2003*; *Everard, 1987*; *Lubke, Everard & Jackson, 1986*). This vegetation type is often dominated by *P. afra* in terms of biomass and cover (*Penzhorn, Robbertse & Olivier, 1974*; *Vlok, Euston-Brown & Cowling, 2003*). These studies, however, focus on large-scale vegetation distribution patterns, and do not take into account localised patterns of community composition. Based on these general descriptions, the restoration of degraded subtropical succulent thicket has primarily used one method: planting *P. afra* to restore vegetation structure and ecological processes (*Mills & Cowling, 2006*; *van der Vyver et al., 2013*).

These planting activities often approached degraded succulent thicket as a uniform vegetation type, planting *P. afra* at an even density across the landscape. This approach failed to account for potential ecological filters and led to highly variable establishment and survival rates, ranging between zero and nearly 100% (*Mills & Robson, 2017*). Subsequent work revealed that much of the poor survival of these plants was a consequence of previously unrecognised environmental factors, including localised frost events resulting

from cold air pooling (*Duker et al., 2020*, *2015*) and herbivory (*van der Vyver et al., 2021*). Soil properties have been suggested as a possible ecological filter responsible for suppressing *P. afra* cover by *Mills et al. (2011)* in some succulent thicket types (however, *P. afra* cover across the study area was relatively low for succulent thicket, and the role of soil pH or plant available macro- and micronutrients was not included in the discussion). The relationship between carbon (C) stock accumulation in restored thicket and soil nutrients has been highlighted (*Mills & Cowling, 2010*) but the actual plant responses are not well understood for *P. afra*.

It is recognised that while subtropical succulent thicket is characterised as being a dense, closed-canopy woodland, a great deal of variation exists within this vegetation and landscape, including the degree of canopy openness and *P. afra* cover (*Vlok, Euston-Brown & Cowling, 2003*), potentially resulting from local edaphic conditions (*Carvalho & Campbell, 2021*; *Mills et al., 2011*). In its most extreme form—calcrete "bontveld"—the thicket plant community occurs as isolated bush clumps restricted to deeper soil pockets in a dwarf shrubland matrix that occurs on shallow soils (*Carvalho & Campbell, 2021*). The bush clumps are largely devoid of typical succulent thicket species, and this represents a local environmental filter (likely high pH) that excludes succulents, such as *P. afra*, that are relatively abundant in immediately adjacent stands of subtropical succulent thicket (figure 2 and table 1 in *Carvalho & Campbell, 2021*). While *Carvalho & Campbell (2021)* attribute the floristic differences between bush clumps and continuous thicket to soil depth and its effect on moisture availability, they did not consider soil chemistry, which is a likely driver for the lack of CAM succulents (such as *P. afra*). Furthermore, *Mills et al. (2011)* also suggest that certain soil properties (electrical conductivity, organic carbon, sand content, and concentrations of total Ca, Zn, and Al) might also cause a decline in *P. afra* abundance and cover, but plant available nutrients were not considered.

The degradation of plant communities can remove evidence of local environmental filters by homogenising the plant community (reduced species diversity, abundance and cover). Thus, instances exist of restoration efforts targeting incorrect local environments (as described previously and in *van der Vyver et al., 2021* and *Duker et al., 2020*). Here we explore the effect of different soils, sourced from historically closed-canopy succulent thicket and an open canopy woodland (Fig. 1), on the establishment of *P. afra* cuttings. We hypothesise that soil chemical properties drive this historical difference in canopy structure (and likely community structure). We find that root growth was hindered in *P. afra* cuttings grown in soils from the historically open canopy woodland, shedding light on how local soil chemical properties may have acted as an environmental filter for succulent species and may impact restoration efforts in this ecosystem.

## METHODS

### Soil sampling

The Sundays River valley (Eastern Cape, South Africa) is characterised by a series of gravelly fluvial terraces formed by the palaeo-Sundays River and its tributaries (*Hattingh, 1994*). The higher terraces lie between 40 and 180 m above the current day Sundays River and their remnants are preserved in the form of calcareous conglomerate capped hills and

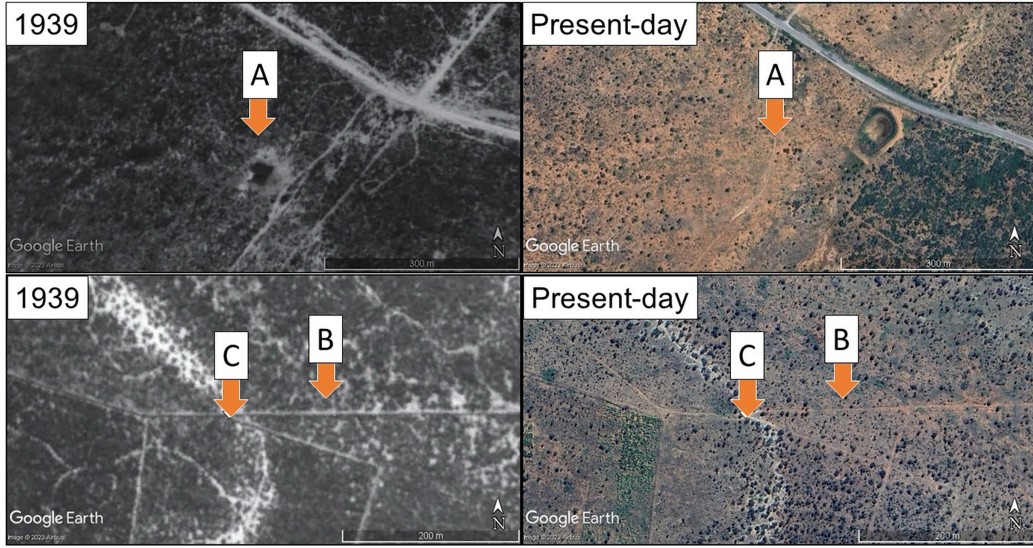

**Figure 1 Past (1939) and present-day aerial imagery of the three locations that soil samples were collected.** Site location indicated by arrows and letters. Site C lacked full canopy cover in 1939, whereas both sites A and B were in intact, closed-canopy succulent thicket. All areas are now open degraded woodlands, with very limited presence of succulent plants. The 1939 images are aerial photographs taken by the South African chief directorate of national geo-spatial information (Reproduced under government's printer authorisation (Authorisation No. 11929 dated 26 March 2025)). Present-day images are satellite images from 2020 (©2021 Google Earth, Maxar Technologies).

ridges (*Hattingh, 1994*). Three sites located on the upper palaeo terraces were identified for soil collection based on the historic degree of vegetation canopy density, as determined from 1939 aerial survey imagery captured by the South African Chief Directorate of National Geospatial Information (Fig. 1, Reproduced under Government's Printer Authorisation). Site A was located on a midslope of an eroded terrace, Site B was located on the lower slope of an eroded tributary terrace and Site C was located 118 m away and 3 m below this on the resistant calcareous capping material of a palaeo-terrace (Fig. 1). At each site, a composite sample was taken from a 5 m² area from 0–20 cm, ensuring that the sample was representative of the local soil conditions affecting *P. afra* establishment. While site-level replication was not included, this design is appropriate for testing how these distinct soil types influence root development under controlled conditions. Soils were tested for carbonates in the field using 10% HCl. Soil properties were assessed at BemLab (Somerset West, South Africa) under standard quality control protocols (see Table S1 for the list of soil properties and values for each soil): air-dried soil samples were sieved to remove gravel and plant roots before analysis; soil pH was measured in a 1M KCl solution (1:2.5 ratio) using an Ohaus Starter 2100 bench pH meter; plant-available phosphorus was determined using the Bray II extraction method, while exchangeable cations ($Mg^{2+}$, $Na^+$, $Ca^{2+}$, $K^+$) and base saturation percentages were measured following 0.2M ammonium acetate extraction (pH 7) and quantified *via* ICP-OES; organic carbon and total carbon content was analyzed using Walkley-Black and dry combustion with a Leco Truspec 1

CHN analyzer, respectively. The soil collected for the experiment (described below) was sifted through a 5 mm sieve to remove any large objects (such as rocks and old roots) and transferred into polypropylene UV-protected plastic seed trays (98-cavity trays with a volume of ~90 cm$^3$ per cavity).

To assess the variation of the soil from the three sites with those found across the study region, we used the soils database associated with the thicket-wide plot experiment (*Mills et al., 2015*; database obtained from Conservation Support Services, Makhanda: https://cssgis.co.za/), where 331 restoration plots were established across the Thicket biome between 2008 and 2009, with various types of soil data collected at each plot. These soils were also analysed at BemLab using the same methods reported above. Here we use the soil results obtained from a surface bulk sample (four samples of ~200 cm$^3$ taken within the plot to a depth of 20 cm) from each plot (101 of 331 plots where all variables were available). The following variables from the thicket-wide plot experiment and the soils used in this study were used to conduct a principle components analysis (PCA): total C, electrical conductivity (mS.m$^{-1}$; 1:5), pH (KCl), P (Bray II), K$^+$, Na$^+$, Ca$^{2+}$ and Mg$^{2+}$. The PCA was conducted using the *prcomp()* function of the *stats* library in R version 4.2.3 (*R Core Team, 2023*).

## Experimental design

Cuttings were harvested from a healthy restored population on shale-derived soils in intact thicket to ensure consistency in plant condition, as previous studies have shown that plant condition affects rooting success (*Potts et al., 2024*). The soil at Site A is likely saline due to recent degradation rather than long-term edaphic processes, making local adaptation to salinity at an ecotypic level unlikely. Similarly, while calcareous soils may have been a persistent substrate, their patchy distribution within a matrix of non-calcareous soils reduces the likelihood of *P. afra* ecotypic adaptation to high-pH conditions.

To minimize potential confounding effects of soil history and plant condition, cuttings were sourced from a single stand with uniform environmental conditions. A total of 588 cuttings (15–20 cm in length)—42 cuttings per plant from 14 plants—were harvested from a single stand (33.47592°S, 25.34329°E) on June 26, 2022. Parent-plant identity for each cutting was tracked throughout the experiment to account for parent-plant effects on root development (*Galuszynski et al., 2023*). All cuttings were standardised to a length of ~15 cm and subsequently planted into ~90 cm$^3$ seed tray cavities labelled with the parent-plant identity and soil type. Moisture content for the parent plants was established by drying three additional cuttings (per plant) at 80 °C for three days—all plants were considered free of water-stress as the moisture content of the cuttings ranged from 83.9–86.9%.

All cuttings were rooted under warm conditions, similar to in-field summer temperatures, in a glasshouse and growth chamber (Conviron model CMP 6010) (Table 1). However, the growth chamber fans were set to the maximum setting to simulate windy, and thus more stressful, conditions. All samples were randomised across seed trays, with an equal representation of parent-plants and soil types in each of the growth conditions. Each cavity was carefully watered (ensuring no spillover between adjacent

**Table 1 Temperature (°C) and relative humidity (%) for the glasshouse and growth chamber during the duration of the experiment.**

|  | Glasshouse | Growth chamber |
| --- | --- | --- |
| Mean average daily temperature (±SD) | 24.7 ± 2.1 | 22.3 ± 0.4 |
| Max mean daily temperature | 29.3 | 22.5 |
| Max temperature | 45.3 | 37.1 |
| Min mean daily temperature | 19.5 | 20.7 |
| Min temperature | 17.5 | 17.6 |
| Mean average daily relative humidity (±SD) | 50.2 ± 8.1 | 55.2 ± 6.3 |

plants) on a weekly basis. To prevent leaching losses, the amount of water added per watering event (100 mL) was limited, ensuring that soil moisture remained below field capacity. This prevented any significant loss of NaCl or other soluble salts, maintaining consistent soil salinity throughout the rooting period.

## Rooting assessment

One cutting from each parent plant per treatment was harvested weekly over seven weeks, from July 11 to September 12, resulting in seven sampling events, with each event yielding a total of fourteen cuttings per treatment. The samples were removed from the planting cavity, and the soil was carefully removed to avoid root damage. Roots were carefully removed using a scalpel, ensuring minimal disturbance to fine structures. After drying at 80 °C for 48 h, residual soil was gently brushed off using soft-bristle brushes to prevent root damage; the roots were not washed with water. The soil-free roots were weighed in grammes to four decimal places.

## Data analysis

All statistical analyses and figures were generated using R version 4.2.3 (*R Core Team, 2023*) with all libraries version control set to 01-Apr-2023 using the *groundhog* library (ver. 3.0.0; *Simonsohn & Gruson, 2023*). The *rstatix* library (ver. 0.7.2; *Kassambara, 2023*) was used for all statistical tests. The following libraries were used in generating figures: *ggplot2* (ver. 3.4.1; *Wickham, 2016*), *ggrepel* (ver. 0.9.3; *Slowikowski, 2021*) and *patchwork* (ver. 1.1.2; *Pedersen, 2023*).

Summary statistics, specifically mean, standard error and boxplots, were used to examine trends of root growth through time. The root dry mass values for each week across treatments were tested for normality and homogeneity of variances using the Shapiro-Wilk Normality Test and Levene's Test, respectively. These tests used the shapiro_test() and levene_test() functions of the *rstatix* library. As many of the tests found non-normal data and heterogeneity of variances, the non-parametric Kruskal-Wallis rank sum test was used to compare for significant differences of root mass amongst root mass values for the combination of soil type and glasshouse or growth chamber. *Post-hoc* Tukey tests were conducted on each site using the Conover-Iman test of multiple comparisons using rank sums (all Kruskal-Wallis tests indicated significant differences). To test for the parent-plant effect (*i.e.*, the impact that the starting condition of the individual from which

cuttings were sourced may have on root development, demonstrated in *Galuszynski et al., 2023*), a Kruskal-Wallis test was conducted of root dry mass grouped by parent plants across all treatments and harvest days.

## RESULTS

Soils from all sites had similar physical properties in terms of texture and water holding capacity (Table S1), but differed in terms of their chemistry. Soils collected from Site C differed substantially to soils collected from Sites A and B. Site C soil reacted strongly with 10% HCl, showing a high $CaCO_3$ content, while soils from Sites A and B, did not react with HCl and therefore lacked substantial carbonate enrichment (*i.e.* <1% $CaCO_3$). The chemical differences amongst the soils from the three sites are shown in the principal components analysis (Fig. 2), which accounted for 61.7% of the variability in the data. Soil from Site C differed from those from Sites A and B in terms of having higher pH and exchangeable $Ca^{2+}$ concentration and a lower plant available P concentration (Figs. 2 and 3, Table S1). Soil C also has a lower salinity and low plant available Fe, Zn and Cu compared to soils from Site A and B. All three soils have intermediate levels of exchangeable Na, Mg, and K (relative to the soils sampled from the thicket-wide plot experiment). Soil from Site A has the lowest pH (6.5, $pH_{KCl}$), is highly saline, and has elevated plant available P and nitrate concentrations (Table S1).

The general trends of root growth through time across the soils in the glasshouse and growth chamber are shown in Fig. 4: cuttings grown in soil from Site C had consistently lower root masses at each harvesting event than those grown in soil from Sites A or B in both the glasshouse and growth chamber. There were significant differences within each harvesting date across the treatments (Fig. 5; all tests: Kruskal-Wallis, $\chi^2_{(5)}$ = 27.3–49.4, $p < 0.0001$, $n = 82$–84). For the majority of harvesting dates, cuttings grown in soil from Site C had significantly lower root mass than those grown in soils from Sites A or B in either growth condition (Fig. 5). The root mass from cuttings grown in soil from Sites A and B did not differ significantly within the same growth condition.

There was a significant difference in root mass amongst cuttings from different parent plants grown in the growth chamber and the glass house (with cuttings lumped together across all the harvest days but separated by growing conditions: Glasshouse: $\chi^2_{(13)}$ = 34.6, $p = 0.00099$, $n = 294$; Growth chamber: $\chi^2_{(13)}$ = 22.0, $p = 0.0559$, $n = 294$; Fig. S1).

## DISCUSSION

The root development of *Portulacaria afra* cuttings grown in the soils from the three different sites and two controlled climate conditions (which simulated warm summer temperatures and regular water availability) revealed that soil chemical properties had a strong effect on root development. Cuttings grown in soil from Sites A and B (sourced from historically closed-canopy succulent thicket) had greater root mass than those grown in soil from Site C (sourced from a historically open woodland), supporting the hypothesis that soil properties can determine where *P. afra*—and, by extension, succulent thicket—can form a closed canopy.

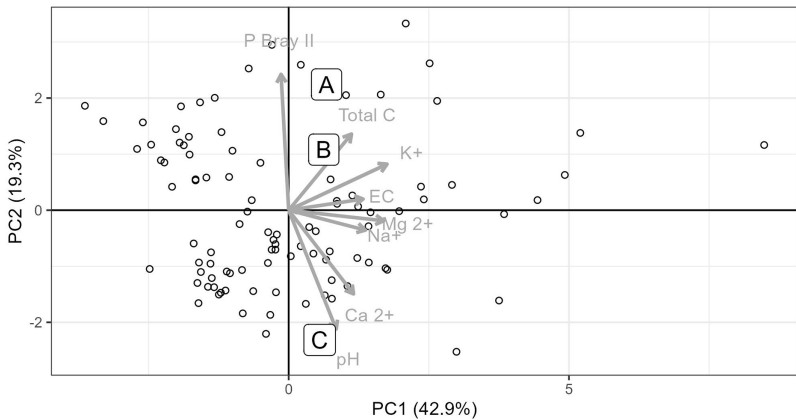

**Figure 2 Principle component analysis of eight soil parameters from 101 thicket-wide plots and the soils from the three sites.** Soils from the three sites (A, B and C) used for the root growth experiments. Soil parameter values for the three soils used in the experiment are reported in Table S1.

While these results indicate that edaphic factors influence *P. afra* establishment, the potential for local adaptation to soil conditions in this species is likely limited. Calcareous soils occur in fragmented patches within the landscape, and the salinity at Sites A and B appears to be a recent consequence of degradation rather than a persistent selective pressure. As such, *P. afra* has likely not developed distinct ecotypes adapted to either saline or calcareous conditions; other common thicket species that co-occur with *P. afra*, such *Pappea capensis* or *Schotia afra* do not show genetic evidence of population isolation based on soil type (*Potts, Hedderson & Cowling, 2013*). Furthermore, because plant condition can influence rooting success (*Potts et al., 2024*), we selected cuttings from a healthy restored population in intact thicket to minimize confounding effects of plant health.

In general, root biomass was lower in the growth chamber, but this was not significant for the majority of matched soils comparisons. Despite being harvested from the same site, individual plant identity did have an effect on root development (as previously demonstrated, *Galuszynski et al., 2023*), with some individuals producing significantly more root mass than others (Fig. S1). This was controlled for in the experimental design and thus, did not impact the outcome of the study (we urge experimental work with *P. afra* cuttings to control for individual plant effects).

Analysis of the soil properties revealed that soils from the paleo-terrace (Site C) differ substantially to soils from Sites A and B. Soils from Site C are highly calcareous with elevated exchangeable Ca and high pH (7.9). The low plant available P concentration (Table S1) is likely related to the presence of soil carbonates due to the formation of insoluble Ca phosphate minerals (*Samadi, 2006*). Phosphate promotes root formation and the low concentrations of bioavailable P could have contributed to the poor root development in cuttings grown in soil from Site C. In addition, the low concentration of bioavailable micronutrients (Fe, Zn and Cu) in Site C soil is also likely attributed to the presence of carbonates and would also have an impact on root development (*Ding, Clode*
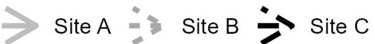

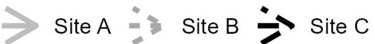

**Figure 3 Properties of soils used in the root growth experiments from the three study sites, contextualized against data from 101 sites across the Albany subtropical thicket biome.** Arrows indicate each of the sites (A, B, C).            
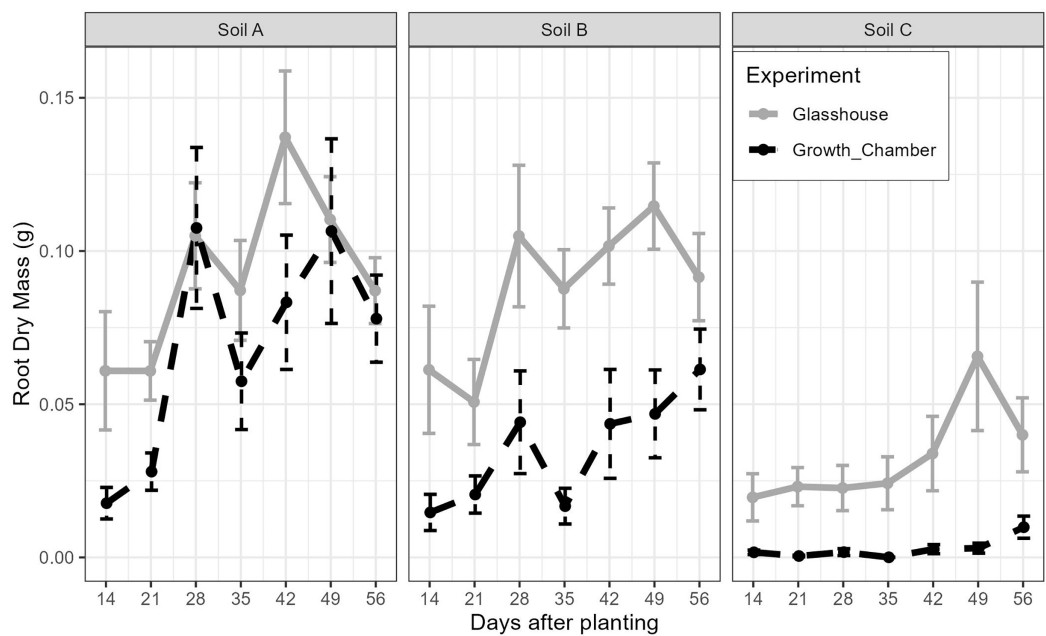

**Figure 4 The mean and standard error of the root dry mass from cutting grown in soil from three sites and growth conditions (glasshouse or growth chamber); *n* = 14 per point.** Black dashed line indicates growth chamber; grey solid line indicates glasshouse.

*& Lambers, 2020*; *Zhao & Wu, 2017*). Soils collected from the terrace slopes (Site A and B), were slightly saline and had a lower pH than soils from Site C. The plant available P concentrations from both Sites A and B were high, especially site B soils which showed an unusually high P concentration for soils that are not under cultivation. In addition, site B soils also had an anomalously high nitrate concentration. The reason for these elevated macronutrients was unclear, but their presence did not appear to significantly increase the root development in comparison to Site A. This suggests that Site A soils are not P limited. Soil from Sites A and B were slightly saline and were higher than the extreme values associated with reduced P. *afra* cover reported by *Mills et al. (2011)*. The results from this study suggest that pH and the presence of carbonates exert a larger effect on root development than salinity. More work needs to be conducted to understand the effect of salinity on root development, given the salinity ranges that occur in semi-arid environments. Some possible mechanisms for the reduced root development in soil from Site C are discussed below, and the potential impact of edaphic conditions is contextualised with respect to vegetation structure and succulent thicket restoration trajectories.

Plant community composition can change rapidly across edaphic transitions, including shifts in dominant plant species and vegetation density (*Acebes et al., 2010*; *Carvalho & Campbell, 2021*; *Moro et al., 2015*; *Siefert et al., 2012*). The Eastern Cape of South Africa provides some dramatic examples of these transitions, with dwarf shrublands (fynbos) and grasslands occurring on sandstone-derived soils, and subtropical thicket occurring on adjacent shale-derived soils under the same climate (*Cowling & Potts, 2015*). These

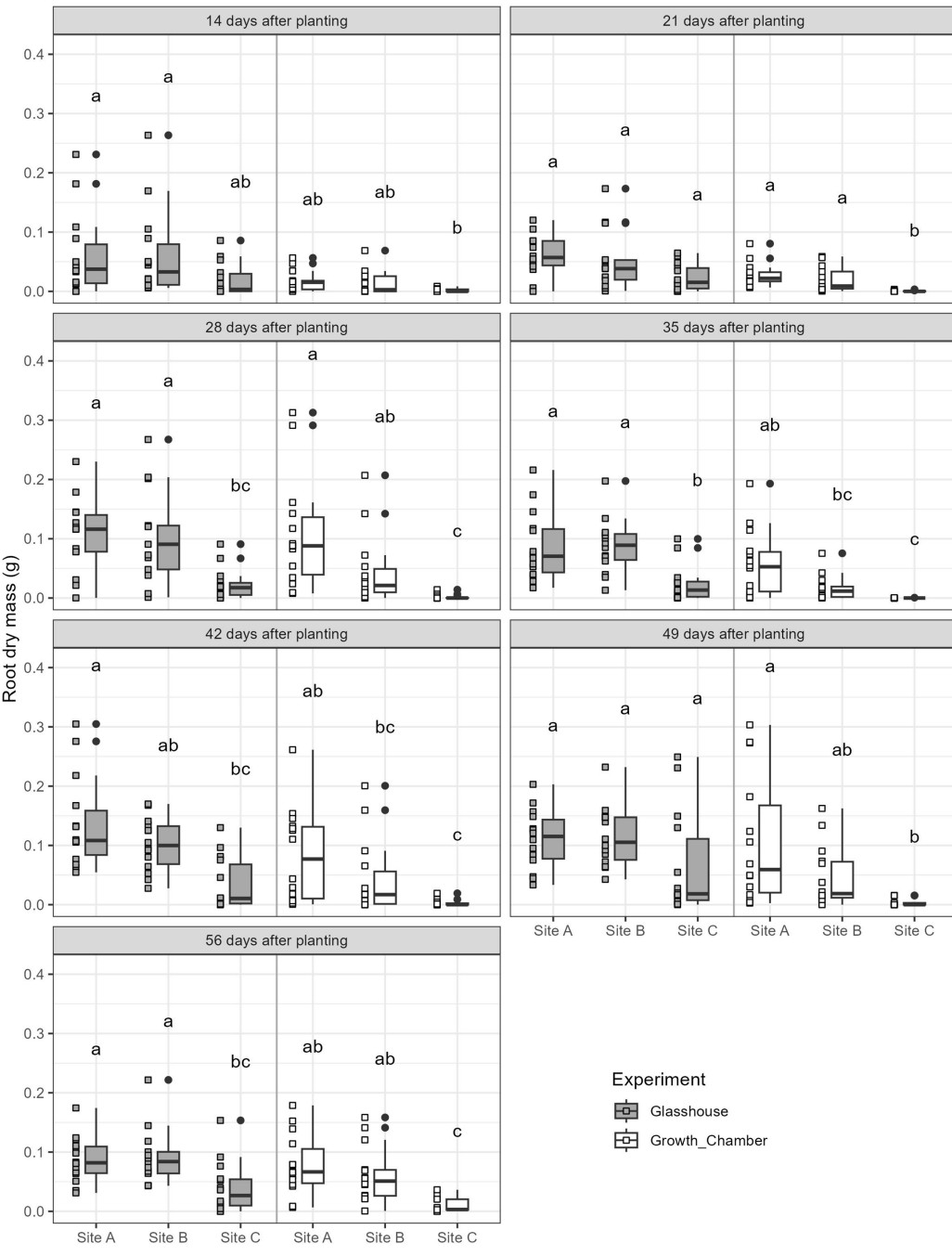

**Figure 5 Boxplots of root dry mass across the seven harvest dates for cuttings grown in soils from three different sites and in the glasshouse or growth chamber.** Plots on the left side of each panel indicate glasshouse, plots on the right side of each panel indicate growth chamber. Dissimilar superscripts denote significant differences among all soils and growth conditions (glass house and growth chamber) at $p < 0.05$ side of each panel. All quantitative data were analysed statistically using Kruskal-Wallis rank sum test and *post-hoc* Conover-Iman rank sum test in R.

vegetation transitions can result from the influence of soil properties on plant establishment, growth rates, and recovery from stresses such as herbivory and drought.

The abundance of *Portulacaria afra* (biomass and cover) in succulent thicket vegetation is partially due to its ability to spread laterally through low-growing branches that root directly into the soil at points of contact (*Stuart-Hill, 1992*). In instances where rooting is impeded by local soil conditions, such as those at Site C in this study (Figs. 2 and 3), this lateral expansion may be limited, and *P. afra* cover may be lower than on adjacent soils (*e.g.*, Sites A and B). Furthermore, while recruitment from seed is poorly understood in this species (and not tested in this study), we postulate that seedling establishment is likely to be limited in less favourable soil conditions. While the processes that inhibit root development in *P. afra* are not explicitly studied here, reduced root development under elevated pH (associated with calcareous soils) has been demonstrated in a variety of woody plant species (*De Klerk, Hanecakova & Jásik, 2008*; *Harbage & Stimart, 1996*; *Holt, Maynard & Johnson, 1998*).

Root initiation in cuttings is dependent on the mobilisation of stored carbohydrates, which provide energy for cell division and differentiation (*Husen & Pal, 2007*). This process of carbohydrate mobilisation is mediated by auxins (*De Almeida et al., 2017*; *Husen, 2008*; *Ruedell, de Almeida & Fett-Neto, 2015*), which are pH-sensitive. When in a high pH environment (above 7), such as that of soil from Site C, auxins are present in an anionic form that is unable to diffuse across cell membranes, preventing the transport of these growth hormones to rooting sites and, thus, potentially inhibiting root development (*Morris, 2000*). Furthermore, soils with high concentrations of carbonates have been demonstrated to impede nutrient uptake and growth in plants (*Ding, Clode & Lambers, 2020*; *Zhao & Wu, 2017*), potentially further reducing root development and disrupting plant productivity.

Poor root development and growth rates may limit lateral spreading of *P. afra* on suboptimal soils and impact this species' resilience to local stressors such as drought and herbivory. Soil pH and nutrient availability can impact plant palatability (*Karki & Goodman, 2011*; *Walker, Marchant & Ethredge, 1975*). While the impact of edaphic properties on the palatability of *P. afra* is currently unknown, herbivory has been speculated to have a strong negative effect on cutting establishment (*van der Vyver et al., 2021*), and over-browsing by domestic livestock has resulted in the near-complete removal of this species across large portions of the biome (*Carvalho, Campbell & du Preez, 2022*; *Lloyd, van den Berg & Palmer, 2002*). This sensitivity to over-browsing, coupled with reduced growth rates due to poor root development and possible nutrient deficiencies on calcareous soils, may contribute to limiting or excluding *P. afra* if more palatable, and thus, more selectively browsed. Moreover, succulent thicket thrives in a region characterized by limited and frequently unpredictable rainfall, where rainfall events can occur at any point throughout the year, and prolonged droughts are a common occurrence (*Archer et al., 2022*; *Mahlalela et al., 2020*; *Vlok, Euston-Brown & Cowling, 2003*). Being a CAM succulent, *P. afra* exhibits remarkable acclimatisation to drought conditions (*Guralnick & Ting, 1986, 1987*; *Bews & Vanderplank, 1930*). Nevertheless, the collective impacts of poor root development, herbivory, and drought may synergistically hinder the persistence of
*P. afra* in calcareous soils, leading to sizable open patches within otherwise dense stands of succulent thicket (Fig. 1).

The results from this study suggest that the low survival of *P. afra* cuttings in some restoration initiatives could be a consequence of local soil properties. Poor establishment and survival of *P. afra* on suboptimal soils (possibly considered a failed restoration effort) may result in open canopy vegetation that reflects a previously unconsidered reference vegetation structure (*Hobbs & Harris, 2001*; *Prach et al., 2019*; *Thorpe & Stanley, 2011*). One potential limitation of this study is the lack of site-level replication, as each soil type was represented by a single composite sample. However, our aim was to assess broad differences in root development across distinct soil types, rather than to capture fine-scale variation within sites. The clear differences in soil properties among sites (Figs. 2 and 3) and the strong response in root development suggest that these edaphic contrasts, rather than site-specific idiosyncrasies, were driving the observed patterns.

The presence of calcrete outcrops, and/or areas of high soil pH, can be fairly common and widespread in the region where succulent thicket ecosystems occur (Fig 2; *Carvalho & Campbell, 2021*). Therefore, succulent thicket exhibiting low vegetation cover or canopy density today may not be an indication of a system once dominated by *P. afra*. In light of this, defining site-specific restoration trajectories is required. Many of these calcrete outcrops can be identified through aerial imagery analysis (Fig. 1) or through landscape elevation profiles which show resistant strata through changing slope angles. Spatial correlation of these features with canopy cover would increase our understanding of the role edaphic properties play in *P. afra* establishment and growth. Such vegetation-soil-terrain relationships are important to inform succulent thicket restoration practices and requires further work.

## CONCLUSION

The soil used as the growing medium strongly affected root initiation and development of *P. afra* cuttings. Cuttings established in soils sourced from a historically open-canopy site with calcareous soils, exhibited reduced root development compared to those grown in soils sourced from now degraded, but historically closed-canopy, thicket sites. This likely reflects an edaphically-induced plant community filtering process in succulent thicket, which has important implications for restoration of this vegetation.

We do not suggest avoiding the reintroduction of *P. afra* in regions with calcareous soils, as it is improbable that this species was entirely absent from these areas (AJ Potts, 2022–2024, personal observations). However, lower establishment and growth rates should be expected in these circumstances. Restoration efforts should acknowledge the natural variation in local conditions, and respond accordingly. Site-specific restoration plans should be developed in collaboration between implementing agents and vegetation scientists to identify suitable planting areas and planting densities, and soil properties should be characterised across areas targeted for restoration.

## ACKNOWLEDGEMENTS

The authors extend their gratitude to Dr. Robbert Duker for selecting the sites and collecting the soils for this project, and to the landowners—Johannes De Lange and Arthur Rudmann—who allowed us to collect soils from their properties. Also to Joel Greaves for providing comments on this document.

### Funding

This work was supported by the National Research Fund of South Africa (Grant Nos. 119379 & 141985) the Natural Resource Management programme of the South African Department of Forestry, Fisheries and the Environment (Project No. E1406). The funders had no role in study design, data collection and analysis, decision to publish, or preparation of the manuscript.

### Grant Disclosures

The following grant information was disclosed by the authors:
National Research Fund of South Africa: 119379 & 141985.
Natural Resource Management Programme of the South African Department of Forestry, Fisheries and the Environment: E1406.

### Competing Interests

Alastair J. Potts is an Academic Editor for PeerJ.

### Author Contributions

- Alastair J. Potts conceived and designed the experiments, analyzed the data, prepared figures and/or tables, authored or reviewed drafts of the article, and approved the final draft.
- Duncan Liddell conceived and designed the experiments, performed the experiments, authored or reviewed drafts of the article, and approved the final draft.
- Catherine E. Clarke analyzed the data, authored or reviewed drafts of the article, and approved the final draft.
- Nicholas C. Galuszynski analyzed the data, authored or reviewed drafts of the article, and approved the final draft.

### Data Availability

The raw data is available in the Supplemental Files.

### Supplemental Information

Supplemental information for this article can be found online at http://dx.doi.org/10.7717/peerj.19303#supplemental-information.

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
