# Peer review of "Restoring South African subtropical succulent thicket using Portulacaria afra: rooting variation across three soil types"

_PeerJ, doi:10.7717/peerj.19303_

## Round 0.1 · original submission · Major Revisions

Ecological restoration in degraded landscapes is a complex and multifaceted process. I think each study that was conducted to improve our perspective on the ecological restoration is very significant. Therefore, I believe your research provides important insights for decision-makers. However, to enhance the clarity of your article, it is important to address certain technical aspects. I strongly encourage a thorough review of the reviewers' suggestions and thoughtful consideration of each recommendation. If you disagree with any suggestion, providing clear and well-reasoned justifications would be beneficial.

Reviewer 1 ·

Basic reporting

A well written article, but the abstract does not reflect the intention of the study. The abstract requires reviewing.

Experimental design

Mostly sound, but researchers did not control for loss of NaCl during the watering process. I am hence not convinced about results of site B.

Validity of the findings

The findings are probably only true when site A and C are contrasted.

·

Basic reporting

No comment

Experimental design

The experimental design appropriate for the answering the research questions. There is one aspect that worries me though. The 588 cuttings from a single stand ((33.47592°S, 25.34329°E). The three sites were:
1. A (slightly-saline)
2. B (non-saline) (in either growth condition) and
3. C (calcareous)
The question is; what were the edaphic conditions under the stand from where the cuttings were harvested. If for example it was harvested from a non-saline site and considering possible plant adaptation to local conditions, is it not possible that material/cuttings harvested from a saline site might do better in soil collected from a site similar to A (slightly-saline) rather than B?

As reviewer I always consider papers from the perspective of a manager. If the above scenario is possible should there not have been a broader scope from where cuttings were collected. The authors might be able to show that no significant differences can be expected. If so, it will have ecological restoration management implications

Validity of the findings

No comment. The findings will clearly contribute to future ecological restoration planning in the thicket.

Additional comments

The research methodology is sound with the one minor "shortcoming" as discussed under "Experimental design" should the authors not be able to show that rooting success is not expected to be impacted by the edaphic conditions where the cuttings were harvested.

The paper is highly relevant to future restoration plan design and methods/approaches.

Reviewer 3 ·

Basic reporting

1) Check your usage of however when joining two clauses

2) You have done a great job in incorporating proper literature and interpreting your results.

3) Your main hypothesis needs a why or a because. You need to justify more why soil should matter.

Experimental design

1) There is a major flaw in your design. You do not have replication at the site level. All the results inferring that there are difference among sites are not valid. You need to have independently measured samples at the site level. All the results suggesting site level differences are not statistically valid. I suggest the following: cluster the sites using the other data from the other plots. Try to place your sites within those clusters. You likely can then pool the replicates from the other plots and make a conclusion that the high pH sites are not suitable for the growth and generalize from site C to the other sites. I have added some R code attached.

2) You make reference to the soil elements in the results without describing them in the methods. This article is therefore not acceptable for publication. How did you measure Fe, Zn and Cu ? How did you extract the cations (Ca, Mg, Na and K)? Ammonium acetate? There are many omissions in the methods. Make sure each result is properly referenced in the methods.

3) How did you remove the roots? Did you wash them? More details are needed here as this is the main result.

Validity of the findings

I believe the author's conclusions. They have reference the literature well and interpreted their findings adequately.

Additional comments

The authors have done an excellent job in writing the introduction and the discussion. They have provided their code and data. There are many positives to the study. The main flaw is that there is not replication at the site level for the soil variables. There are also results described that are not in the methods. These two major flaws detract from the great writing and discussion.


library(tidyverse)
dat <- read_csv("peerj-110882-twp_soils.csv")
tmp <- dat %>%
filter(str_detect(string = Plot,pattern = "Site"))
ggplot(dat,aes(x=pH_KCl,y=PBrayII))+
geom_point()+
geom_point(data=dat[dat$Plot %in% tmp$Plot,],col="red",size=5)
###
#Run a cluster analyses here on pH and bray II

#Assign your sites two these (Hopefully) two clusters

#run a comparison using the two clusters to make proper inference using borrowed replicates
t.test(pH ~ cluster)
#

---

## Round 0.2 · accepted · Accept

I would like to thank you for accepting the referees' suggestions and improving your article based on their suggestions. Your article is ready to publish. We look forward to your next article.

·

Basic reporting

No further comments

Experimental design

Authors have addressed my concerns adequately.

Validity of the findings

My comments stay the same as my initial review. It is a highly significant article for restoration planning of Eastern Cape Thicket Restoration.

Additional comments

No additional comemnts